# GAZE FOLLOWING IN QUESTION ANSWERING: A COMPREHENSIVE BENCHMARK FOR VISION-LANGUAGE MODELS

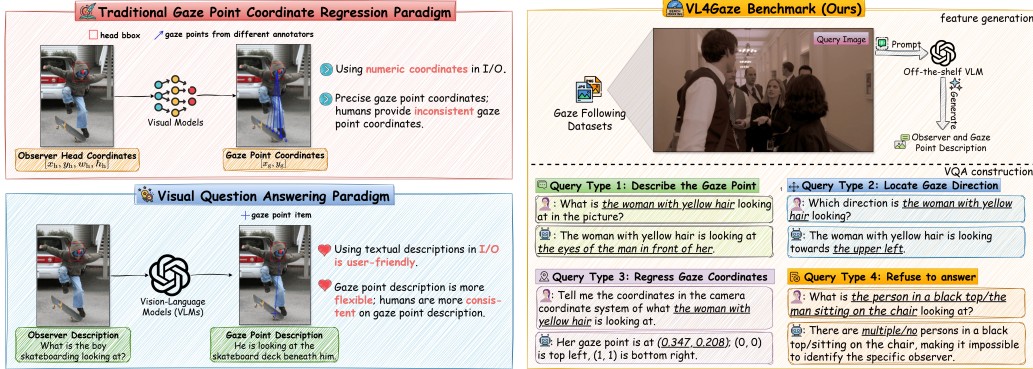

(a) Gaze following in VQA paradigm  (b) GazeVQA benchmark

Figure 1: Conventional gaze following methods take scene images and observer information as input to estimate the observed object or point of regard. However, describing gaze using coordinate positions is neither intuitive nor accurate. Actually, people often describe the gaze target using language instead. Thus motivated, we introduce the first large-scale dataset to explore the capabilities of vision-language models (VLMs) for the gaze following task. Our dataset consists of 410K text-image pairs and includes four query types. Building upon this dataset, we establish the first benchmark for VLM-based gaze following, unlocking the potential of VLMs and paving the way for future research in this area.

## ABSTRACT

Gaze following aims to infer human intention within scene images. Conventional methods typically rely on scene and face images to regress the gaze point coordinates which is unnatural and restrictive. Recently, vision-language models (VLMs) have attracted significant attention for their powerful reasoning abilities, raising an important question: can VLMs be leveraged to advance the gaze following? In this work, we introduce GazeVQA, the first large-scale text-image dataset for VLM-based gaze following. GazeVQA is the first to provide accurate textual annotations for both observers and gaze targets, along with natural language question-answering (QA) pairs tailored for the gaze following task. The dataset contains 410K QA pairs across 102K scene images, offering rich supervision for training and evaluating VLMs. Building on GazeVQA, we establish the first benchmark for VLM-based gaze following. Experiments demonstrate that existing VLMS exhibit limited zero-shot performance on gaze following. However, with training on our dataset, their performance improves significantly, demonstrating the potential of GazeVQA to drive progress in this area. We will release the dataset and code to facilitate future research.

## 1 INTRODUCTION

Gaze behavior is a fundamental part of human non-verbal communication (Just & Carpenter, 1980; Tonini et al., 2023). Understanding and analyzing gaze with computer vision techniques is key to

Table 1: Comparison between GazeVQA and existing gaze-following datasets. The 410K QA pairs in GazeVQA are generated from a single sampling pass. Thanks to our diverse pools of descriptions and QA templates, the dataset supports multi-round sampling for enhanced linguistic diversity.

| Dataset | Annotations | | Query Types | | | | Dataset Scale |
|---|---|---|---|---|---|---|---|
| | Observer | Gaze Target | Description | Direction | Coordinate | In/Out | |
| GazeFollow (Recasens et al., 2015) | Bbox | Points | ✗ | ✗ | ✓ | ✓ | 130K |
| VideoAttentionTarget (Chong et al., 2020) | Bbox | Points | ✗ | ✗ | ✓ | ✓ | 164K |
| GOO (Tomas et al., 2021) | Bbox | Points & Object | ✗ | ✗ | ✓ | ✗ | 172K |
| ChildPlay (Tafasca et al., 2023b) | Bbox | Points & Class | ✗ | ✗ | ✓ | ✓ | 258K |
| **GazeVQA (Ours)** | **Text** | **Text** | ✓ | ✓ | ✓ | ✓ | **410K** |

understanding human attention, intention and future actions (Fathi et al., 2012; Wei et al., 2018). As a core task, gaze following aims to predict the visual target that a person is looking at in a given scene image (Recasens et al., 2015), which has wide applications in many fields, including human-computer interaction (Admoni & Scassellati, 2017), neuroscience (Tafasca et al., 2023a), social psychology (Capozzi et al., 2019), *etc.*

Recent advanced datasets and network architectures in gaze following (Hu et al., 2023; Ryan et al., 2025; Chong et al., 2020; Tafasca et al., 2024) have commonly adopted a paradigm where the observer's head coordinates serve as input and gaze point coordinates as output. Although performance has improved, this paradigm has inherent limitations. First, it is not user-friendly, as both input and output are numeric coordinates, making it hard for users to naturally formulate inputs and interpret outputs. Second, precise gaze point regression is overly rigid and not well-suited for describing the gaze following task. In fact, even humans struggle to pinpoint exact gaze coordinates in complex scenes, so obtaining descriptive labels of gaze regions is often more practical. As illustrated in Figure 1 (a), the blue arrows represent the gaze point coordinates annotated by different annotators, which vary significantly, *e.g.*, the maximum difference along the x-axis approaches one-third of the image width. However, all annotators actually refer to the same object (*i.e.,* "skateboard deck").

To tackle the aforementioned challenges, in contrast to the typical regression paradigm for estimating gaze targets, we address the gaze following task in a new perspective by designing a visual question answering paradigm, which can offer a more natural, reasonable, and user-friendly solution. As shown in Figure 1, compared to traditional models that output a single value to represent the gaze location, in the VQA paradigm, people can ask the model a question like "What is the boy skateboarding looking at?" and receive a more understandable answer, such as "He is looking at the skateboard deck beneath him". Such visual-language interactive paradigm is more friendly for downstream tasks like understanding and action prediction.

However, it is challenging to construct a high-quality gaze following benchmark within the VQA framework. Thanks to the success of Vision-Language Models (VLMs) across a variety of tasks, such as image captioning (Luu et al., 2024) and human-object interaction detection (Lei et al., 2024), we introduce GazeVQA, a large-scale benchmark constructed under the VQA paradigm, to fully unlock the potential of VLMs in gaze following.

As illustrated in Figure 1(b), GazeVQA systematically covers four core types of gaze-related questions that reflect how humans naturally perceive and express gaze behavior: (1) **Describe the gaze point**: the most intuitive and commonly used formulation, which captures the semantic content of the gaze target and allows models to associate gaze with meaningful objects or regions; (2) **Locate the direction**: designed to reflect humans' coarse-level perception of gaze orientation and to help models learn directional cues from eye appearance and head pose; (3) **Regress gaze point coordinates**: aligned with coordinate-based paradigms, suitable for scenarios requiring precise numerical predictions by mapping visual cues to spatial positions; (4) **Refuse to answer**: introduced to improve model robustness by prompting the model to identify and reject ill-posed queries, such as those referring to multiple or nonexistent individuals. Note that if the observer's **gaze falls outside the image frame, our system automatically detects this condition and returns "gaze outside the frame"**, and consequently would not provide an in-image description or precise coordinate output. The GazeVQA benchmark is automatically constructed on top of standard gaze-following datasets using a VLM-powered pipeline, yielding nearly 410K question–answer pairs across 102K images.

Table 1 highlights the comparison between GazeVQA and existing benchmarks. Based on the constructed the GazeVQA benchmark, we fine-tune multiple VLMs and develop evaluation protocols tailored to each question type. Our experimental results demonstrate significant gains over existing baselines, showing that GazeVQA not only improves the practical utility of gaze-following systems but also endows VLMs with a deeper and more natural understanding of human gaze behavior.

In summary, this study introduces GazeVQA, a large-scale benchmark designed to explore the gaze-following capabilities of VLMs under a VQA paradigm. GazeVQA provides rich and diverse QA pairs that focus on modeling fine-grained eye cues, bridging the gap between human-centric visual understanding and language reasoning. By fine-tuning VLMs on GazeVQA, we demonstrate significant performance improvements across multiple gaze-related tasks, highlighting the potential of language-guided supervision for advancing gaze analysis.

## 2 RELATED WORK

### 2.1 GAZE FOLLOWING

The gaze following task was introduced by Recasens et al. (Recasens et al., 2015), aiming to predict a person's gaze point in an image as a 2D coordinate. Their method employed a two-branch architecture, where one branch captures scene-level context and the other analyzes the cropped head to estimate gaze direction. The fused features are then used for final coordinate prediction. Subsequent works enhanced this design by incorporating cues such as depth (Fang et al., 2021; Gupta et al., 2022), body pose (Gupta et al., 2022), 3D head orientation (Horanyi et al., 2023), and temporal or contextual information (Chong et al., 2020; Gupta et al., 2024). More recently, Ryan et al. (Ryan et al., 2025) departed from the two-branch architecture and achieved strong performance using frozen vision foundation encoders with lightweight decoders, demonstrating the potential of foundation models in this task. Despite these advances, existing methods largely follow a coordinate-based input/output paradigm. To our knowledge, no prior work has explored gaze following under the visual question answering paradigm using vision-language models.

### 2.2 VISION-LANGUAGE MODELS

In recent years, VLMs (Bai et al., 2025; Grattafiori et al., 2024) have demonstrated strong capabilities in contextual understanding within VQA tasks. This progress is largely attributed to the breakthroughs of Large Language Models (LLMs) (Bai et al., 2023; Touvron et al., 2023; Jiang et al., 2024) in language modeling, as well as the outstanding performance of modern vision encoders (Liu et al., 2021; Caron et al., 2021; He et al., 2022) in visual feature extraction. However, gaze behavior, a crucial cue in human non-verbal communication, remains underexplored in current VLMs. While VLMs possess a degree of contextual understanding and can seemingly be used to predict gaze direction, they still struggle to interpret gaze behavior accurately in less aligned scenarios. This is largely due to the lack of fine-grained modeling of eye details and their spatial relationship to the gaze target. Such limitations hinder their potential to analyze human attention, infer intent, and anticipate future actions. To address this gap, we focus on enhancing VLMs' ability to perceive and understand gaze itself by introducing a novel benchmark called GazeVQA.

## 3 PRELIMINARY STUDY

VLMs have recently demonstrated strong generalization capabilities in image understanding. Gaze following, however, is a more fine-grained sub-task of image understanding that requires attention to subtle visual cues, particularly in the eye region of the observer. Despite the rapid progress in VLMs, their effectiveness on the gaze following task remains unexplored. In this section, we first investigate the zero-shot performance of VLMs on the gaze following task. We evaluate two VLMs, including Qwen2.5-VL-7B-Instruct (Bai et al., 2025) and GPT-4o (Achiam et al., 2023). For each case, we provide the image and the corresponding question to the model and record the generated answer. We show the qualitative examples in Figure 2.

Our results show that while VLMs can correctly infer the gaze target in simple scenarios, they often fail in more complex situations, such as scenes involving intricate social relationships or when the

gaze direction is not aligned with head orientation. These experiments suggest that although VLMs can leverage contextual and social cues, they struggle to capture fine-grained eye features and precise spatial geometry, which are critical for accurate gaze following. This finding also highlights the need for a large-scale and accurately annotated gaze following benchmark tailored to VLMs, motivating the development of our GazeVQA dataset.

# 4 GAZEVQA BENCHMARK

To unleash the potential of VLMs, we introduce GazeVQA, the first large-scale text-image benchmark for gaze following. The benchmark contains 410K question-answer (QA) pairs across 102K scene images, providing rich supervision for training and evaluating vision-language models. Below, we describe the process of constructing the benchmark in detail.

## 4.1 DATASET CONSTRUCTION

To enhance the gaze understanding capability of VLMs, we construct the GazeVQA benchmark based on traditional gaze-following dataset GazeFollow (Recasens et al., 2015). We leverage its collected images along with annotations of **head bounding boxes** and **gaze point coordinates** (or cases where the gaze is outside the image) to automatically generate detailed question-answer (QA) pairs. As shown in Figure 3, GazeVQA is built through three key steps: extracting observer features using an off-the-shelf VLM (GPT-4o (Achiam et al., 2023) or Qwen-VL series (Bai et al., 2025) in practice), extracting gaze point features also with the same VLM, and constructing VQA samples by combining the obtained features. In the following sections, we elaborate on each step and provide a comprehensive analysis of the constructed dataset.

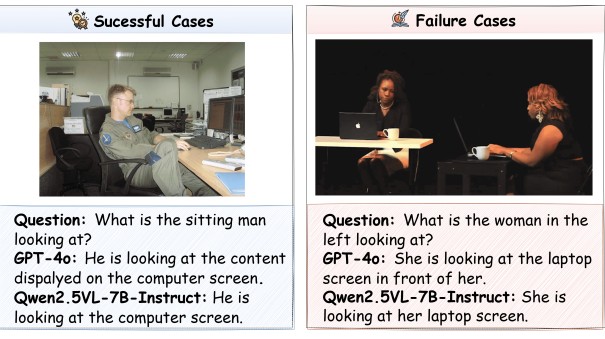

Figure 2: Zero-shot VLM performance in gaze following.

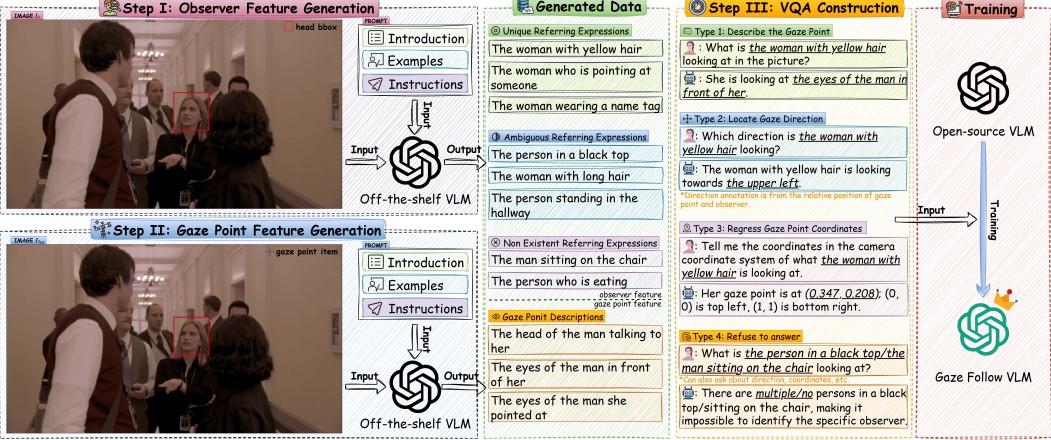

Figure 3: Overall construction pipeline of GazeVQA benchmark.

### 4.1.1 STEP I: OBSERVER FEATURE GENERATION

In the VQA paradigm of gaze following, users typically inquire about various types of gaze information by referring to a specific person within the image using natural language descriptions. To simulate this human behavior during the construction of our dataset, we employ an off-the-shelf VLM $\pi$ to automatically generate descriptive expressions for the target person. Specifically, for an

image $I \in \mathbb{D}$ from a gaze following dataset, we input the image $I_\mathrm{h}$, which includes the head bounding box annotated for a specific observer, into a VLM to generate observer feature description about that person:

$$\pi(\mathcal{C}_\mathrm{o} \oplus I_\mathrm{h}) \Rightarrow \mathcal{E}, \tag{1}$$

where $\mathcal{C}_\mathrm{o}$ denotes the prompt used for observer feature generation; $\mathcal{E} \equiv \{\mathcal{E}_\mathrm{u}, \mathcal{E}_\mathrm{a}, \mathcal{E}_\mathrm{n}, \mathcal{E}_\mathrm{s}\}$ represents the observer feature descriptions generated by the VLM, including **u**nique referring expressions $\mathcal{E}_\mathrm{u}$, **a**mbiguous referring expressions $\mathcal{E}_\mathrm{a}$, **n**on-existent referring expressions $\mathcal{E}_\mathrm{n}$, and **s**ubject pronouns $\mathcal{E}_\mathrm{s}$ (*e.g.*, "he" or "she"). Notably, each expression in the unique referring set $\mathcal{E}_\mathrm{u}$ captures a different descriptive view of the same observer. We detail the structured prompt design of $\mathcal{C}_\mathrm{o}$ in Appendix.

Among the generated descriptions $\mathcal{E}$, the unique referring expressions $\mathcal{E}_\mathrm{u}$ characterize the observer with visual cues that allow for uniquely and unambiguous identification, and are later used to construct answerable VQA data. In contrast, ambiguous referring expressions $\mathcal{E}_\mathrm{a}$ contain ambiguous descriptions that may correspond to multiple individuals in the scene, thus failing to uniquely localize the target. Non-existent referring expressions $\mathcal{E}_\mathrm{n}$, on the other hand, describe individuals who do not appear in the image at all. These latter two types of expressions are intentionally included to build declined-response cases in our VQA dataset.

### 4.1.2 STEP II: GAZE POINT FEATURE GENERATION

Distinguishing from the traditional gaze following paradigm, which is entirely based on gaze point coordinates, we introduce textual descriptions of gaze points in GazeVQA for constructing VQA data. Specifically, for samples where the gaze point lies within the image, we annotate each instance with both the observer and their corresponding gaze location, resulting in $I_\mathrm{hg}$. We then utilize the VLM $\pi$ to generate feature descriptions of objects, people, landscapes, *etc.*, at the gaze point specified in $I_\mathrm{hg}$ by the observer:

$$\pi(\mathcal{C}_\mathrm{g} \oplus I_\mathrm{hg}) \Rightarrow \mathcal{E}_\mathrm{g}, \tag{2}$$

where $\mathcal{C}_\mathrm{g}$ denotes the prompt used for gaze point feature generation. $\mathcal{E}_\mathrm{g}$ is the gaze point description generated by the VLM, comprising multiple expressions that each capture a different descriptive angle of the same gaze point. These expressions encourage relational descriptions with respect to the observer and refer to the observer using personal pronouns, *e.g.,* "the head of the man *talking to her*" rather than "the head of the man talking to the woman". We present the specific structured prompt design of $\mathcal{C}_\mathrm{g}$ in Appendix.

### 4.1.3 STEP III: VISUAL QUESTION ANSWERING CONSTRUCTION

After obtaining the observer referring expressions $\mathcal{E}_\mathrm{u}$ and the gaze point referring expressions $\mathcal{E}_\mathrm{g}$, we construct corresponding VQA pairs to form the GazeVQA benchmark, which aims to enhance the gaze-awareness capabilities of VLMs. GazeVQA supports four types of questions, each generated through a multi-stage sampling process: an observer referring expression $e_u \sim \mathcal{E}_\mathrm{u}$ is first sampled from the observer referring set, followed by a gaze point referring expression $e_g \sim \mathcal{E}_\mathrm{g}$ from the gaze point referring set. Finally, a question template is sampled from the semantically diverse template set $\mathbb{T}$, into which the sampled expressions and subject pronouns $\mathcal{E}_\mathrm{s}$ are filled to generate a complete question-answer (QA) pair. This multi-stage sampling procedure ensures both linguistic diversity and rich content in the generated QA pairs.

1. **Describe the gaze point**: This type prompts a description of the objects, people, or regions located at the gaze point of a specific observer identified by $e_u$. A target description $e_g$ is first sampled, followed by a prompt template $\mathcal{T}_\mathrm{desc} \sim \mathbb{T}_\mathrm{desc}$. The QA pair is then generated as:

$$(\mathcal{C}_\mathrm{gaze}, \mathcal{E}_\mathrm{gaze}) = \begin{cases} \mathcal{T}_\mathrm{desc}(e_u, e_g, \mathcal{E}_\mathrm{s}), & \text{if gaze inside,} \\ \mathcal{T}_\mathrm{desc}(e_u, \text{out}, \mathcal{E}_\mathrm{s}), & \text{if gaze outside.} \end{cases} \tag{3}$$

2. **Locate gaze direction**: This type asks about the directional region in which a specific observer $e_u$ is looking. The $360°$ space around the head is divided into eight angular regions, and the region label $d$ is computed accordingly. A template $\mathcal{T}_\mathrm{dir} \sim \mathbb{T}_\mathrm{dir}$ is then used to produce the QA pair:

$$(\mathcal{C}_\mathrm{gaze}, \mathcal{E}_\mathrm{gaze}) = \mathcal{T}_\mathrm{dir}(e_u, d, \mathcal{E}_\mathrm{s}). \tag{4}$$

3. **Regress gaze point coordinates**: This type queries the normalized 2D coordinates of the gaze point in the camera coordinate system, for an observer identified by $e_u \sim \mathcal{E}_u$. Unlike traditional approaches that rely on bounding boxes (Gupta et al., 2022; Horanyi et al., 2023; Gupta et al., 2024), the observer is specified using natural language descriptions $e_u$. Specifically, a coordinate template $\mathcal{T}_{\text{coord}} \sim \mathbb{T}_{\text{coord}}$ is sampled, and the coordinates $\mathbf{c}$ are defined as:

$$\mathbf{c} = \begin{cases} (x, y), & \text{if gaze inside image,} \\ (-1, -1), & \text{if gaze outside image.} \end{cases} \quad (5)$$

The QA pair is then formed as:

$$(\mathcal{C}_{\text{gaze}}, \mathcal{E}_{\text{gaze}}) = \mathcal{T}_{\text{coord}}(e_u, \mathbf{c}, \mathcal{E}_s). \quad (6)$$

4. **Refuse to answer**: For ambiguous or invalid observer references $e_a, e_n$ sampled from $\mathcal{E}_a$ or $\mathcal{E}_n$, a refusal response is generated. A template $\mathcal{T}_{\text{refuse}} \sim \mathbb{T}_{\text{refuse}}$ is selected based on the error type, and the QA pair is produced as:

$$(\mathcal{C}_{\text{gaze}}, \mathcal{E}_{\text{gaze}}) = \mathcal{T}_{\text{refuse}}(e_a/e_n, \mathcal{E}_s). \quad (7)$$

In summary, each question–answer instance in GazeVQA is represented as:

$$(\mathcal{C}_{\text{gaze}}, I, \mathcal{E}_{\text{gaze}}) \sim \mathbb{D}_{\text{gaze}}, \quad (8)$$

where $\mathcal{C}_{\text{gaze}}$ is the generated question, $I$ is the associated image or context, and $\mathcal{E}_{\text{gaze}}$ is the expected answer. The combination of rich template libraries and expressive sampling strategies ensures linguistic variety, semantic flexibility, and strong data augmentation. Detailed templates $\mathbb{T}$ are provided in Appendix.

### 4.1.4 ANALYSIS

In constructing GazeVQA-Bench, we adopt Qwen-VL-Max (Bai et al., 2025) as the $\pi$ model, using Chinese as the prompt language, and retain the original structure and official train/test splits of the GazeFollow (Recasens et al., 2015) dataset. Based on this, after filtering out a small number of samples for which the VLM consistently fails to generate a valid $\mathcal{E}$ or $\mathcal{E}_g$ format and performing a single sampling pass, we obtain the statistics shown in Table 2. The training set comprises approximately 98K images and yields around 390K QA pairs, while the test set includes about 5K images and 19K QA pairs.

Table 2: Comprehensive Statistics of GazeVQA Dataset

| Statistic | Type 1 | Type 2 | Type 3 | Type 4 |
|---|---|---|---|---|
| images (train/test) | | 97,615 / 4,782 | | |
| QA pairs (train/test) | | 390,460 / 19,128 | | |
| Avg. Q length (chars) | 23.31 | 23.62 | 64.00 | 16.93 |
| Avg. A length (chars) | 26.12 | 12.24 | 40.61 | 40.75 |
| Avg. Diff. sampling | 23.93 | 6.68 | 13.36 | 6.68 |

We will release all QA pairs used during training and testing, along with intermediate representations to facilitate further data augmentation, including observer feature descriptions $\mathcal{E} \equiv \{\mathcal{E}_u, \mathcal{E}_a, \mathcal{E}_n, \mathcal{E}_s\}$ from Step I, gaze point description $\mathcal{E}_g$ from Step II, and the template $\mathbb{T}$ used in Step III to generate QA pairs. Thus, researchers can also easily expand the dataset as needed. The average number of maximum distinct sampling passes per data type is summarized in Table 2.

### 4.2 SUPERVISED TRAINING OF VLMs USING GAZEVQA

Ultimately, we employ GazeVQA to train an off-the-shelf VLM $\pi_\theta$ to enhance its gaze tracking performance. We conduct standard supervised fine-tuning on $\pi_\theta$, with the loss function defined as follows:

$$\mathcal{L} = -\mathbb{E}_{(\mathcal{C}_{\text{gaze}}, I, \mathcal{E}_{\text{gaze}}) \sim \mathbb{D}_{\text{gaze}}} \log P(\mathcal{E}_{\text{gaze}} | \mathcal{C}_{\text{gaze}}, I; \theta), \quad (9)$$

where the trained VLM $\pi_\theta$ is parameterized by $\theta$. We enhance its capabilities in person localization, eye gaze awareness, and gaze point regression through supervised training on the GazeVQA benchmark, thereby optimizing the performance of the VLM in gaze following.

It is noteworthy that GazeVQA conforms to the standard VQA format of VLMs, which enables its integration into the large-scale post-training process of VLMs (Ziegler et al., 2019; Ouyang et al., 2022; Swamy et al., 2025). We utilize VLM-generated data along with annotated information from gaze following datasets to construct four types of questions in GazeVQA. However, we can still leverage this information to create more types of questions (*e.g.*, inquiring about where the person in the head bounding box is looking) or other output formats (*e.g.*, preference pair-based data).

Table 3: Comparison between the best-performing commercial VLMs under prompt-based answer formatting rules and smaller-scale open-source VLMs trained with supervision on GazeVQA. † indicates that, due to the high cost of commercial API usage, evaluation was performed on a random sample of 1K QA pairs from our GazeVQA test set. The best and second best are denoted as blue and orange. These results demonstrate that our small-scale VLMs trained on GazeVQA significantly outperform larger commercial VLMs in gaze understanding tasks.

| Methods | Training | Describe gaze point | | Locate gaze direction | | Regress gaze coordinates | | Refuse to answer |
|---|---|---|---|---|---|---|---|---|
| | | BLEU↑ | ROGUE-L↑ | Angle Error↓ | Term Match↑ | L2↓ | Acc$_{in/out}$↑ | Acc$_{ra}$↑ |
| GPT-41† | Prompt | 24.17 | 45.94 | 45.54 | 0.633 | 0.158 | 0.988 | 0.711 |
| GPT-4o† | | 15.62 | 37.01 | 45.54 | 0.633 | 0.183 | 0.771 | 0.908 |
| Claude4-Opus† | | 26.87 | 48.98 | 28.17 | 0.750 | 0.185 | 1.000 | 0.928 |
| Gemini-2.5-pro† | | 26.60 | 51.01 | 29.21 | 0.748 | 0.127 | 0.952 | 0.839 |
| Qwen-VL-Max | | 18.58 | 39.95 | 29.12 | 0.728 | 0.142 | 0.564 | 0.762 |
| Qwen2.5-VL-72B-Instruct | | 15.95 | 36.64 | 29.69 | 0.729 | 0.134 | 0.503 | 0.779 |
| LLaMA-3.2-11B-Vision | GazeVQA | 58.98 | 66.16 | 16.08 | 0.851 | 0.097 | 0.973 | 1.000 |
| Qwen2.5-VL-7B-Instruct | | 59.54 | 66.69 | 15.31 | 0.856 | 0.101 | 0.974 | 1.000 |

Table 4: Parameter sensitivity analysis for Qwen models on GazeVQA. We analyze the impact of model size, number of test-time generations, and data sampling times during supervised training. The best best are denoted as blue. These results demonstrate that increasing the model size, performing multiple sampling passes during testing, and increasing the sampling frequency from GazeVQA during training can all lead to further performance improvements. Users can make their own trade-offs between efficiency and performance based on specific needs.

| Parameter | Value | Describe gaze point | | Locate gaze point | | Regress gaze coordinates | | Refuse to answer |
|---|---|---|---|---|---|---|---|---|
| | | BLEU↑ | ROUGE-L↑ | Angle Error↓ | Term Match↑ | L2↓ | Acc$_{in/out}$↑ | Acc$_{ra}$↑ |
| Model size | 3B | 57.26 | 65.71 | 19.17 | 0.820 | 0.112 | 0.974 | 1.000 |
| | 7B | 59.54 | 66.69 | 15.31 | 0.856 | 0.101 | 0.974 | 1.000 |
| | 32B | 60.61 | 66.95 | 13.78 | 0.872 | 0.086 | 0.969 | 1.000 |
| Best of N | 1 | 59.54 | 66.69 | 15.31 | 0.856 | 0.101 | 0.974 | 1.000 |
| | 4 | 75.43 | 76.20 | 5.80 | 0.946 | 0.055 | 0.992 | 1.000 |
| | 8 | 80.68 | 80.09 | 3.96 | 0.963 | 0.040 | 0.994 | 1.000 |
| | 16 | 84.80 | 82.84 | 3.02 | 0.972 | 0.032 | 0.997 | 1.000 |
| Data sampling times | 1x | 58.32 | 65.45 | 17.68 | 0.834 | 0.110 | 0.967 | 1.000 |
| | 2x | 59.54 | 66.69 | 15.31 | 0.856 | 0.101 | 0.974 | 1.000 |
| | 4x | 60.59 | 66.38 | 13.99 | 0.871 | 0.090 | 0.963 | 1.000 |

# 5 EXPERIMENTS

## 5.1 EXPERIMENTAL SETTINGS

**Baseline Methods.** We conduct a comprehensive evaluation using representative VLMs from different model families, covering a range of parameter scales and training strategies. Specifically, for proprietary models, we include mainstream commercial systems such as GPT-41, GPT-4o (Achiam et al., 2023), Claude4-Opus (Anthropic, 2024), Gemini-2.5-pro (Team et al., 2023), and Qwen-VL-Max (Bai et al., 2025). For open-source models, we evaluate the strong-performing Qwen2.5-VL-72B-Instruct (Bai et al., 2025). Since these models lack training specifically tailored for the gaze-following task, we introduce *prompt-based answer formatting rules* to ensure a fair comparison with our fine-tuned models. The example of the prompt rule template will be shown in Appendix.

**Implementation Details.** In our experiments, we focus on fine-tuning relatively lightweight models from two representative families of VLMs, namely Qwen (Bai et al., 2025) and LLaMA (Grattafiori et al., 2024), specifically Qwen2.5-VL-7B-Instruct and LLaMA-3.2-11B-Vision-Instruct. These models are supervised fine-tuned on our proposed GazeVQA dataset and evaluated against stronger baseline methods to highlight the necessity and value of the benchmark in advancing gaze understanding within VLMs. To enhance data diversity, we perform two rounds of sampling on the training set and train the model for one epoch on it. Additional training hyperparameters and implementation details are provided in Appendix.

**Evaluation metrics.** Using our GazeVQA test set, we evaluate models on four gaze understanding tasks: For Type 1 (Describe gaze point), we assess description quality using BLEU (Papineni et al., 2002) and ROUGE-L (Lin & Och, 2004); for Type 2 (Locate gaze direction), we compute angular error (0-180°) with directional mapping and direction term matching scores (shared characters/max length); for Type 3 (Regress gaze coordinates), we report L2 distance (Recasens et al., 2015) and gaze in/out of image classification accuracy; for Type 4 (Refuse to answer), where the questions

Table 5: **Generalization evaluation** on the VideoAttentionTarget test set with models trained on the GazeFollow portion of GazeVQA. The best best are denoted as blue. These results indicate that, even under domain generalization settings, VLMs trained on our GazeVQA dataset significantly outperform currently larger-scale VLMs.

| Methods | Training | Describe gaze point | | Locate gaze direction | | Regress gaze coordinates | | Refuse to answer |
|---|---|---|---|---|---|---|---|---|
| | | BLEU↑ | ROUGE-L↑ | Angle Error↓ | Term Match↑ | L2↓ | Acc$_{in/out}$↑ | Acc$_{ra}$↑ |
| Qwen-VL-Max | Prompt | 28.40 | 46.78 | 36.62 | 0.680 | 0.225 | 0.601 | 0.694 |
| Qwen2.5-VL-72B-Instruct | | 29.31 | 46.71 | 39.59 | 0.656 | 0.219 | 0.597 | 0.726 |
| Qwen2.5-VL-32B-Instruct | GazeVQA | **34.86** | **54.95** | **32.96** | **0.718** | **0.175** | **0.738** | **1.000** |

Table 6: Ablation study on the effect of question formulation strategies. The best best are denoted as blue. These results show that training with all QA types from GazeVQA outperforms using only description-type queries, confirming the mutual benefits among different task types and validating the design of our benchmark. Moreover, performance can be further improved by incorporating bounding boxes as additional question inputs to serve as data augmentation.

| Methods | Describe gaze point | | Locate gaze direction | | Regress gaze coordinates | | Refuse to answer |
|---|---|---|---|---|---|---|---|
| | BLEU↑ | ROUGE-L↑ | Angle Error↓ | Term Match↑ | L2↓ | Acc$_{in/out}$↑ | Acc$_{ra}$↑ |
| w/o supervised training | 26.94 | 54.10 | 83.05 | 0.392 | 0.339 | 0.593 | 0.971 |
| training only on description-type queries | 58.96 | 65.68 | 37.02 | 0.685 | 0.410 | 0.754 | 0.390 |
| + bounding box as additional input | **59.79** | **66.87** | 15.17 | **0.860** | **0.094** | 0.972 | 1.000 |
| Qwen2.5-VL-7B-Instruct (GazeVQA) | 59.54 | 66.69 | **15.31** | 0.856 | 0.101 | **0.974** | 1.000 |

themselves are problematic, we measure refusal accuracy. All metrics employ optimal prediction-reference matching for multi-annotation data.

## 5.2 MAIN RESULTS

Table 3 presents a comprehensive comparison between the best-performing commercial VLMs and our supervised small-scale models trained on GazeVQA. Overall, GazeVQA-supervised models significantly outperform commercial ones across nearly all metrics. Our fine-tuned Qwen2.5-VL-7B-Instruct achieves the best performance on 5 of 7 metrics. Compared to Claude4-Opus, it improves BLEU from 26.87 to 59.54 (+121.6%) and ROUGE-L from 48.98 to 66.69 (+36.2%) in Type 1; reduces angle error from 28.17° to 15.31° (–45.7%) and raises direction term match from 0.750 to 0.856 (+14.1%) in Type 2; decreases L2 error from 0.127 to 0.101 (–20.5%) and improves in/out classification accuracy from 0.952 to 0.974 (+2.3%) over Gemini-2.5-pro in Type 3; and achieves perfect rejection (1.000) in Type 4. These results validate the effectiveness of GazeVQA supervision in enabling superior gaze understanding compared to larger commercial models.

## 5.3 PARAMETER SENSITIVITY ANALYSIS

Table 4 presents our parameter sensitivity analysis on GazeVQA. First, increasing the **model size** from 3B to 32B consistently improves performance across all tasks, indicating that larger models better capture complex gaze semantics. Moreover, increasing the **Best of N** value leads to notable gains, highlighting the multi-modal nature of our predictions and aligning with the inherent variability of human gaze. Finally, increasing the number of **data sampling times** during training demonstrates that, following the GazeVQA sampling strategy, moderate augmentation (e.g., 2×) effectively improves performance, enabling users to balance accuracy and efficiency according to their needs.

## 5.4 GENERALIZATION EVALUATION

To assess the generalization ability of our models trained on GazeVQA, we construct a new test set from the VideoAttentionTarget (Chong et al., 2020) dataset. Following the same QA construction strategy as used for GazeVQA, we generate QA pairs from VideoAttentionTarget and randomly sample 20K QA pairs as test set.

As shown in Table 5, our supervised model outperforms larger-scale VLMs across all metrics in generalization evaluation. It improves BLEU by 18.9% and ROUGE-L by 17.6% in Type 1, reduces angle error by 16.7% and increases direction term match by 9.5% in Type 2, lowers L2 distance by 20.1% and boosts in/out-of-image accuracy by 23.6% in Type 3, and raises refusal accuracy from

0.726 to 1.000 in Type 4. These results confirm the robustness of our model and domain-invariant gaze understanding.

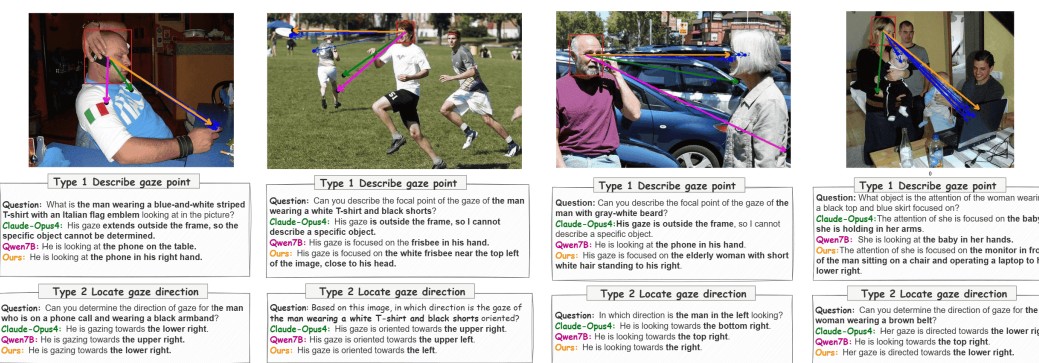

Figure 4: Visualization results of gaze following from three VLMs: the best commercial model (Claude4-Opus), the baseline (Qwen2.5-VL-7B-Instruct), and our supervised GazeVQA-trained model (Ours). Type 1 and 2 QA are shown in text. For Type 3, predicted gaze points are arrows unless outside image. Blue arrows mark multi-human ground truth.

## 5.5 ABLATION STUDY

Table 6 presents an ablation on question types in GazeVQA. The **w/o supervised training** baseline (Qwen2.5-VL-7B-Instruct with prompt-only) performs much worse than our fine-tuned model, confirming GazeVQA's necessity for supervised gaze understanding. Training **only on description-type queries** (Type 1) leads to weaker results even on description metrics, showing that diverse question types mutually enhance learning. Incorporating **bounding box as additional input** further boosts performance, demonstrating flexibility of our VQA-based training.

## 5.6 VISUALIZATION RESULTS

As illustrated in Figure 4, the visualization results clearly demonstrate the superiority of our supervised trained model compared to both the original baseline model (Qwen2.5-VL-7B-Instruct) and the best-performing commercial VLM (Claude4-Opus). Although the commercial model exhibits a certain capability in gaze recognition and outperforms the baseline to some extent, it still faces challenges in capturing fine-grained gaze cues. In contrast, our fine-tuned model delivers more accurate predictions that align better with human annotations across all three types of gaze tasks.

## 6 CONCLUSION

In this work, we propose GazeVQA, the first large-scale benchmark designed to explore the potential of VLMs for the gaze following task. Unlike conventional coordinate regression paradigms, our approach reformulates gaze following as a VQA problem, enabling more natural, interpretable, and user-friendly interaction. GazeVQA introduces accurate textual annotations for both observers and gaze targets, and covers four core question types: describe gaze point, locate gaze direction, regress gaze coordinates and refuse to answer. This design enables the dataset to support a diverse range of use cases. Constructed via a scalable VLM-powered pipeline, the dataset comprises 410K question-answer pairs across 102K scene images. Experimental results show that existing VLMs perform suboptimally in gaze following tasks, but their performance can be significantly improved through supervised training on GazeVQA, confirming the effectiveness of our proposed benchmark.

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

# A  QUESTION ANSWERING CONSTRUCTION TEMPLATE

This section details the template structures $\mathbb{T}$ used to construct the GazeVQA benchmark. The templates follow a multi-stage sampling process to generate linguistically diverse question-answer pairs, as described in Step III of Section 3.1 in the main paper.

TYPE 1: DESCRIBE THE GAZE POINT

This template set $\mathcal{T}_{\text{desc}} \sim \mathbb{T}_{\text{desc}}$ is used to query the object or region at which an observer is looking. It employs a conditional structure based on whether the gaze point is inside or outside the image frame.

$$(\mathcal{C}_{\text{gaze}}, \mathcal{E}_{\text{s}}) = \begin{cases} \mathcal{T}_{\text{desc}}(e_u, e_g, \mathcal{E}_{\text{s}}) & \text{if gaze inside} \\ \mathcal{T}_{\text{desc}}(e_u, \text{out}, \mathcal{E}_{\text{s}}) & \text{if gaze outside} \end{cases} \tag{10}$$

**Observer Reference Processing**: To align with natural human tendencies during question answering, the observer reference $\mathcal{E}_{\text{s}}$ is transformed into $\mathcal{E}_{\text{s}}'$ through the following process:

- Personal pronouns (*he/she*) are used with 70% probability.

- Descriptive noun phrases (e.g., "the woman in red") are used with 30% probability.

- Pronouns and phrases are contextually adapted, e.g., *he* → *his*, *him*; "the woman in red" → "the woman in red's.".

**Gaze Point Description Templates $\mathbb{T}_{\text{desc}}$:**

- **Question:** What is $e_u$ looking at?
  *Gaze Inside Answer:* $\mathcal{E}_{\text{s}}'$ is looking at $e_{\text{g}}$.
  *Gaze Outside Answer:* We cannot see what $\mathcal{E}_{\text{s}}'$ is looking at specifically because the gaze extends outside the frame.

- **Question:** Can you describe the focal point of the gaze of $e_u$?
  *Gaze Inside Answer:* $\mathcal{E}_{\text{s}}'$ gaze is focused on $e_{\text{g}}$.
  *Gaze Outside Answer:* $\mathcal{E}_{\text{s}}'$ gaze is outside the frame, so I cannot describe a specific object.

- **Question:** What object is the attention of $e_u$ focused on?
  *Gaze Inside Answer:* The attention of $\mathcal{E}_{\text{s}}'$ is focused on $e_{\text{g}}$.
  *Gaze Outside Answer:* The attention of $\mathcal{E}_{\text{s}}'$ is directed outside the image, so the specific object cannot be determined.

**Placeholder Definitions**:

- $e_u$: Observer referring expression (e.g., "the person in blue")

- $e_g$: Gaze point referring expression (e.g., "the bubble machine in his hand")

- $\mathcal{E}_{\text{s}}'$: Context-adapted observer reference (e.g., "he"/"his")

TYPE 2: LOCATE GAZE DIRECTION

This template set $\mathcal{T}_{\text{dir}} \sim \mathbb{T}_{\text{dir}}$ queries the direction of the observer's gaze, which is quantized into eight 45-degree regions around the head.

**Locate Gaze Direction Templates $\mathbb{T}_{\text{dir}}$:**

$$(\mathcal{C}_{\text{gaze}}, \mathcal{E}_{\text{gaze}}) = \mathcal{T}_{\text{dir}}(e_u, d, \mathcal{E}_{\text{s}}) \qquad (11)$$

- **Question:** In which direction is $e_u$ looking?
  *Answer:* $\mathcal{E}_{\text{s}}'$ is looking towards the $d$.

- **Question:** Can you determine the direction of gaze for $e_u$?
  *Answer:* $\mathcal{E}_{\text{s}}'$ is gazing towards the $d$.

- **Question:** Based on this image, in which direction is $e_u$'s gaze oriented?
  *Answer:* Based on this image, $\mathcal{E}_{\text{s}}'$ gaze is oriented towards the $d$.

**Placeholder Definitions:**

- $e_u$: Observer unique referring expression (e.g., "the boy on the left")
- $d$: Gaze direction (e.g., "left")
- $\mathcal{E}_{\text{s}}'$: Context-adapted observer reference (e.g., "he"/"his")

TYPE 3: REGRESS GAZE COORDINATES

This template set $\mathcal{T}_{\text{coord}} \sim \mathbb{T}_{\text{coord}}$ requests the precise image coordinates of the gaze point. The responses follow a normalized coordinate system.

$$\mathbf{c} = \begin{cases} (x, y) & \text{if gaze inside} \\ (-1, -1) & \text{if gaze outside} \end{cases} \qquad (12)$$

$$(\mathcal{C}_{\text{gaze}}, \mathcal{E}_{\text{gaze}}) = \mathcal{T}_{\text{coord}}(e_u, \mathbf{c}, \mathcal{E}_{\text{s}}) \qquad (13)$$

**Gaze Point Coordinates Template $\mathbb{T}_{\text{coord}}$**

- **Question:** Can you tell me the image coordinates where $e_u$ is looking?
  *Gaze Inside Answer:* Certainly, the position where $e_u$ is looking is at coordinates $c$, using the (x,y) format where (0,0) represents the top-left corner of the image and (1,1) represents the bottom-right corner in this normalized coordinate system.
  *Gaze Outside Answer:* I'm sorry, the specific coordinates where $\mathcal{E}_{\text{s}}'$ is looking extend beyond the image boundaries.

- **Question:** What are the specific image coordinates where $e_u$ is looking?
  *Gaze Inside Answer:* The specific coordinates where $\mathcal{E}_{\text{s}}'$ is looking are $c$.
  *Gaze Outside Answer:* It appears the gaze of $\mathcal{E}_{\text{s}}'$ extends beyond the frame edges.

- **Question:** Please indicate the image coordinates of the gaze point of $e_u$.
  *Gaze Inside Answer:* The image coordinates of the gaze point of $\mathcal{E}_{\text{s}}'$ are $c$.
  *Gaze Outside Answer:* The gaze point of $\mathcal{E}_{\text{s}}'$ falls outside the image boundaries.

- **Question:** Can you tell me the image coordinates of the gaze point of $e_u$? Output should be in (x,y) format, where (0,0) is top-left and (1,1) is bottom-right, with x and y rounded to three decimal places; if outside the image, output (-1,-1).
  *Gaze Inside Answer:* $c$.
  *Gaze Outside Answer:* (-1,-1).

- **Question:** What are the specific image coordinates where $e_u$ is looking? Output in (x,y) format, where (0,0) is top-left and (1,1) is bottom-right, with x,y as floats rounded to three decimal places; if outside image output (-1,-1).
  *Gaze Inside Answer:* $c$.
  *Gaze Outside Answer:* (-1,-1).

- **Question:** Please indicate the image coordinates of the gaze point of $e_u$. Requirements: 1. Output must be normalized coordinates (x,y) → values between 0–1, rounded to three decimal places, with top-left as origin. 2. If outside image, output (-1,-1). 3. Output only the coordinates themselves.
  *Gaze Inside Answer:* $c$.
  *Gaze Outside Answer:* (-1,-1).

During testing, to facilitate coordinate extraction, we only use the last three templates that directly output the coordinates.

**Placeholder Definitions**:

- $e_u$: Observer unique referring expression (e.g., "the woman in the left")

- $c$: Normalized gaze coordinates in (x.xxx, y.yyy) format

- $\mathcal{E}'_s$: Context-adapted observer reference (e.g., "he"/"his")

TYPE 4: REFUSE TO ANSWER

This template set $\mathcal{T}_{\text{refuse}} \sim \mathbb{T}_{\text{refuse}}$ handles cases where the observer reference is ambiguous or non-existent. These templates ensure the model can properly handle invalid queries.

$$(\mathcal{C}_{\text{gaze}}, \mathcal{E}_{\text{gaze}}) = \mathcal{T}_{\text{refuse}}(e_a/e_n, \mathcal{E}_s) \tag{14}$$

**Refuse to Answer Templates** $\mathbb{T}_{\text{refuse}}$**:** Randomly choose between ambiguous ($e_a$) and non-existent ($e_n$) observer scenarios. The corresponding $\mathbb{T}_{\text{refuse}}$ includes:

**Ambiguous Observer (multiple candidates)**

- **Question:** What is $e_a$ looking at in the image?
  *Answer:* There are multiple people matching the description of $e_a$, unable to determine the specific gaze object. Please provide a more specific description and try again.

- **Question:** In which direction is $e_a$ gazing?
  *Answer:* Sorry, the description of $e_a$ is not unique in the image. Multiple people may match. Please provide more detailed appearance or location information.

- **Question:** Can you tell me the image coordinates of $e_a$'s gaze point?
  *Answer:* Since $e_a$ may refer to multiple people, it is difficult to determine the gaze coordinates. Please specify a clearer object.

**Non-Existent Observer (no matching person)**

- **Question:** In which direction is $e_n$ looking in the image?
  *Answer:* There is no person matching the description of $e_n$ in the image. Please confirm if the description is correct or modify and try again.

- **Question:** What is $e_n$ looking at?
  *Answer:* No object matching $e_n$ was detected. Unable to provide gaze content. Please correct the description.

- **Question:** What are the image coordinates of $e_n$'s gaze point?
  *Answer:* There is no individual matching $e_n$ in the image. Therefore, unable to obtain gaze coordinates. Please modify the description and try again.

**Placeholder Definitions:**

- $e_a$: Observer ambiguous referring expression (e.g., "the standing man")

- $e_n$: Observer non-existent referring expression (e.g., "the girl in red dress")

DISTINCT SAMPLING ENABLED BY GAZEVQA

Questions are generated through the following multi-stage sampling process:

1. Sampling of observer expressions ($e_u \sim \mathcal{E}_u$)
2. Sampling of gaze point expressions ($e_g \sim \mathcal{E}_g$), when applicable
3. Selection of question templates ($\mathcal{T} \sim \mathbb{T}$)

This sampling process promotes linguistic diversity by ensuring that, even for the same visual input, the generated questions vary in both phrasing and structure. As a result, it enables flexible and scalable dataset expansion. The average and maximum number of distinct sampling passes per data type are analyzed in the Section 3.1 (Analysis) of the main paper.

## B  EXPERIMENTAL SETTINGS

PROMPT-BASED ANSWER FORMATTING RULES

Since the baseline models lack training specifically tailored for the gaze-following task, we introduce *prompt-based answer formatting rules* to ensure a fair comparison with our fine-tuned models. Below, we provide one example instruction for each template set, randomly selected from each type, along with its original question prompt.

**Type 1: Describe the Gaze Point**
*Original question:* What is $e_u$ looking at in the picture?
*Answer formatting instruction:* What is $e_u$ looking at in the picture? If the person's gaze is inside the image, respond like: "He is looking at the bubble machine in his hand." If the gaze is outside the image, respond like: "We cannot see what he is looking at specifically because the gaze extends outside the frame. Please describe $e_u$'s gaze target based on this format."

**Type 2: Locate Gaze Direction**
*Original question:* In which direction is $e_u$ looking?
*Answer formatting instruction:* In which direction is $e_u$ looking? Answer using only the direction of gaze in the camera coordinate system, relative to the head position of the described person (i.e., top, bottom, left, right). Use the format: "He is looking towards the lower left." to answer from the perspective of standing on the right side of $e_u$'s gaze direction. Do not provide any further explanation.

**Type 3: Regress Gaze Coordinates**
*Original question:* Can you tell me the image coordinates of the gaze point of eu? Output should be in (x,y) format, where (0,0) is top-left and (1,1) is bottom-right, with x and y rounded to three decimal places; if outside the image, output (-1,-1).
*Answer formatting instruction:* Can you tell me the image coordinates of the gaze point of $e_u$? Output should be in the format (x,y), where (0,0) is the top-left and (1,1) is the bottom-right of the image, with x and y rounded to three decimal places. If the gaze point is outside the image, output (-1,-1). Output only the normalized (x,y) coordinates with three decimal places, for example, (0.123,0.456). If the gaze point is outside the image, respond with (-1,-1). Do not output any other text. Please output the gaze coordinates of $e_u$: (x,y) =

**Type 4: Refuse to Answer**
*Original question:* In which direction is $e_n$ looking in the image?
*Answer formatting instruction:* In which direction is $e_n$ looking in the image? If exactly one person matches the description, answer normally, for example: "He is looking at the bubble machine in his hand." Otherwise, respond with: "There is no person matching the description of $e_n$ in the image. Please confirm if the description is correct or modify and try again." Please provide the gaze content for $e_n$:

| Phase | Hyperparameter | Value | Remark |
|---|---|---|---|
| **GazeVQA Construction** | temperature | 0.5 | - |
| | top p | 0.9 | - |
| | top k | 25 | - |
| | data sampling times | 2 | augment scale in VQA construction |
| **Supervised Training** | optimizer | AdamW | - |
| | learning rate | 1e-6 | - |
| | epoch | 1 | - |
| | batch size | 192 | - |
| | lr scheduler | cosine | - |
| | warmup ratio | 0.1 | - |
| | image max pixels | 262144 | limited pixels to prevent out of memory |

Table 7: Hyperparameter configuration in experiments.

## B.1 IMPLEMENTATION DETAILS

We primarily utilize the PyTorch[1], vllm[2], and Transformers[3] libraries to implement our method, while employing DeepSpeed[4] for multi-GPU parallel training. All our experiments are conducted on a single machine with 8 A800 80GB SXM GPUs. We strive to use consistent irrelevant parameters in both the main experiments and ablation studies to ensure fairness and consistency in comparison. All critical hyperparameters settings are displayed in Table 7.

We utilize paid APIs provided by Alibaba Cloud, Google, OpenAI, and Anthropic to obtain experimental and evaluation results related to models such as Qwen-VL-Max (Bai et al., 2025), Gemini-2.5-pro (Team et al., 2023), GPT-41, GPT-4o (Achiam et al., 2023), and Claude4-Opus (Anthropic, 2024). The multiple LLMs applied in our experiments use the latest versions as of July 1, 2025, specifically Qwen-VL-Max[5], gemini-2.5-pro-06-17[6], gpt-41-0414-global[7], gpt-4o-1120-global[8], and claude-opus4[9].

## C USE OF LARGE LANGUAGE MODELS

We used a large language model (OpenAI GPT) solely for writing assistance, including polishing grammar, improving readability, and refining the presentation of the manuscript. The model was not used for ideation, experimental design, analysis, or generation of scientific content. All technical contributions and experiments are entirely our own.

## D PROMPT AND INSTRUCTIONS

In this section, we present the prompt design for observer and gaze point feature generation.

### D.1 OBSERVER FEATURE GENERATION

In Table 8, we present the prompts for observer feature generation, which are designed to produce features describing the appearance, location, and other characteristics of the observer specified by the head bounding box in the image. The prompts are structured as follows:

1. *Preamble* – An introduction describing the observer feature generation task
2. *Examples* – JSON-formatted examples of the generated features
3. *Instructions* – Requirements that the generated content must adhere to

---

[1] https://github.com/pytorch/pytorch
[2] https://github.com/vllm-project/vllm
[3] https://github.com/huggingface/transformers
[4] https://github.com/microsoft/DeepSpeed
[5] https://qwenlm.github.io/blog/qwen-vl
[6] https://deepmind.google/models/gemini/pro
[7] https://openai.com/index/gpt-4-1
[8] https://platform.openai.com/docs/models
[9] https://www.anthropic.com/claude/opus

4. *Ending* – Ending text to prompt the LLM

## D.2 GAZE POINT FEATURE GENERATION

We present in Table 9 the prompt utilized for gaze point feature generation, aimed at producing gaze point features for a specified observer within an image. The prompt is structured as follows:

1. *Preamble* - An introduction describing the task of gaze point feature generation
2. *Examples* - JSON format examples of feature generation
3. *Instructions* - Guidelines or requirements to be followed during content generation
4. *Ending* - Ending text to prompt the LLM

Table 8: Prompt for observer feature generation.

| Section | Content |
|---------|---------|
| Preamble | `<Task Description>` Provide natural language descriptions for the person marked by a yellow bounding box (bbox_person). Descriptions must be based solely on the clean image (without any bbox markers). Count all clearly visible people in the image and output a JSON with five fields: 1. boxed_person_head_center: "(x, y)" with x and y normalized between 0-1, top-left=(0, 0), bottom-right=(1, 1), three decimal places. 2. total_persons: Total number of clearly visible people, including bbox_person. 3. bbox_person_sex: "male" or "female" based on the bbox_person's gender. 4. unique_referring_expressions: An object with five subfields that uniquely identify bbox_person: • appearance: Describe hair color, ethnicity, gender, age group, ... • clothing: Describe color/style of tops/bottoms and visible accessories ... • action: Describe body/leg posture (standing, sitting, etc.), excluding gaze direction ... • position: Describe bbox_person's absolute location in the image or a specific scene ... • combination: Select any number of features from the above categories (appearance, clothing, action, position\|especially including position and clothing color) to create 1-2 combined descriptions ... 5. ambiguous_referring_expressions: When total_persons > 1, select the most conspicuous other person in the image (besides the bbox_person) and output 1 shared feature description\|for example, ... 6. non_existent_expressions: A list containing exactly 1 randomly generated phrase that does not apply to any person in the image. |

| Section | Content |
|---|---|
| Examples | `<Examples>`
`Example 1:`
`Output JSON:`
`{`
  `"boxed_person_head_center":  "(0.860, 0.422)",`
  `"total_persons":  4,`
  `"bbox_person_sex":  "male",`
  `"unique_referring_expressions":  {`
   `"appearance":  null,`
   `"clothing":  "the man in a green shirt",`
   `"action":  null,`
   `"absolute_position":  "the man sitting on the right side`
`of the green sofa",`
   `"combination":  ["the man in a green shirt sitting on`
`the right side of the green sofa"]`
  `},`
  `"ambiguous_referring_expressions":  ["the sitting man"],`
  `"non_existent_expressions":  ["the sitting little girl"]`
`}`

`Example 2:`
`...` |
| Instructions | `<Requirements>`
`1.  Do not output any text outside JSON, and do not describe`
`the bounding box.`
`2.  The description must be fluent and natural, conform`
`to everyday conversational habits, and must include a noun`
`phrase referring to the bbox_person such as "the man" or "the`
`little girl".`
`3.  The "unique_referring_expressions" must unambiguously`
`identify the bbox_person.  If features in following types:`
`"appearance," "clothing," or "action" is visually ambiguous`
`or can not significantly distinguish the bbox_person from`
`others in the image, confidently output null.  The penalty`
`for mistakenly predicting null is relatively small, but`
`including ambiguous or non-unique features will incur a much`
`greater penalty.`
`4.  If total_persons is 1, then "ambiguous_referring_expressions"`
`should be an empty array [].  if the total number of people >`
`1, ...` |
| Ending | `<Output>`
`{`
  `"boxed_person_head_center":  "(0.258, 0.607)",`
`Please continue from the given part of the JSON above to`
`complete the full JSON format (including the already provided`
`part) and output it.` |

Table 9: Prompt for gaze point feature generation.

| Section | Content |
|---------|---------|
| Preamble | `<Task Description>` The image contains a person whose head is marked by a red bounding box (the boxed person) and a point marked at the orange cross center (the cross point). The cross point is actually the gaze point of the manually annotated bounding box person. Therefore, its x, y coordinates are absolutely accurate. However, considering that multiple semantic objects at different depths may overlap at the same coordinates, further semantic localization should be performed by integrating depth information with the following reference details:
• The cross point cannot be on the boxed person's own head, back, or neck\|areas they cannot see themselves.
• The cross point is more likely to be on prominent objects or items related to the boxed person's actions, like another person's face or objects held in hands.
• The cross point cannot be behind the direction the boxed person's head facing.

Depending on their relationship, the cross point falls into one of three categories. For each category, describe the cross point as follows:
• Category 1 – Cross point directly related to the boxed person. It is:
1. a body part of the boxed person.
2. a non-human object the boxed person is leaning on or riding.
3. a non-human object held, touched, pointed at by the hands, or stepped on/kicked by the feet of the boxed person.
Description format: pronoun (him/her) based on the boxed person's gender + relationship phrase + visual/semantic description.
Examples: "his own hand", "the horse she is riding".
• Category 2 – Cross point directly related to an unboxed person (anyone else in the image). If Category 1 does not apply, then the cross point is:
1. a body part of an unboxed person(considering head and hand first).
2. a non-human object the unboxed person is leaning on or riding.
3. a non-human object held, touched, pointed at by the hands, or stepped on/kicked by the feet of an unboxed person.
Description format: unique description of the unboxed person (relative position to the boxed person, absolute position in the image->such as: far left (0.194, 0.183) first from the right in the second row, interaction with the boxed person, or distinctive appearance) + relationship phrase + visual/semantic description. Always refer to the boxed person as "him"/"her".
Examples: "the girl in red holding hands with him on his right side".
• Category 3 – Cross point not directly related to any person. If neither Category 1 nor 2 apply, the cross point is usually scenery or a placed object.
Description format: pronoun (him/her) + spatial relation to the boxed person OR absolute image position + visual/semantic description.
Examples: "the bushes to his left", "the red car on the right side of the image". |

| Section | Content |
|---|---|
| Examples | `<Examples>`
Return a JSON object with the following fields:
1. "boxed_person_head_center": The boxed person head center's normalized coordinates.
2. "cross_point": The cross point's normalized coordinates.
3. "relationship_category": One of "Category 1 – Cross point directly related to the boxed person", "Category 2 – Cross point directly related to an unboxed person", and "Category 3 – Cross point not directly related to any person".
4. "cross_point_descriptions": 1–3 diverse, fluent, concise descriptions following the category's format.
Coordinate Format: "(x, y)" with x and y normalized between 0–1, top-left=(0, 0), bottom-right=(1, 1), three decimal places.
Example 1:
{
  "boxed_person_head_center": "(0.550, 0.422)",
  "cross_point": "(0.655, 0.546)",
  "relationship_category": "Category 1 – Cross point directly related to the boxed person",
  "cross_point_descriptions": ["the tip of the chalk he is holding touching the blackboard", "the letter he is writing with chalk on the blackboard", "the chalk in his hand pressing against the board"]
}

Example 2:
... |
| Instructions | `<Strict Output Constraints>`
1. Descriptions must follow the specified format for each category and provide clear visual(color + semantic) + spatial cues(detailed location, such as the second person on the right side of the image) to uniquely identify the area indicated by the crosshair in the image, with particular emphasis on specifying its semantics.
2. Spatial references must use camera/image coordinates (image left = left, image front = toward camera).
3. In descriptions, refer to the boxed person only as "he" or "she"\|do not use "boxed person", "cross point", or other terms.
4. The description must directly point to the cross point.
5. No explanatory text outside JSON. The output language must be English. |
| Ending | `<Output>`
{
  "boxed_person_head_center": "(0.357, 0.173)",
  "cross_point": "(0.194, 0.183)",
Please continue from the given part of the JSON above to complete the full JSON format (including the already provided part) and output it. |