# OpenReview forum: "Gaze Following in Question Answering: A Comprehensive Benchmark for Vision-Language Models"
_ICLR.cc/2026/Conference — ICLR 2026 Conference Withdrawn Submission_

### Official Review · Reviewer_gL9r · 2025-10-22

**Soundness:** 2
**Presentation:** 2
**Contribution:** 2
**Rating:** 4
**Confidence:** 4

**Summary:**

This paper focuses on gaze following in the context of vision-language models (VLMs). To leverage VLMs to advance gaze following, the authors introduce a large-scale text-image dataset named GazeVQA for VLM-based gaze following. The GazeVQA dataset contains 410K question-answer pairs across 102K scene images, supporting the training and evaluation of VLMs. Various VLMs are evaluated on the GazeVQA benchmark.

**Strengths:**

1. The authors propose a large-scale text-image dataset named GazeVQA for VLM-based gaze following, which is a novel and interesting setting.

2. The authors proposed 4 types of QA to evaluate the capabilities of VLMs in gaze following.

3. This paper is well organized and easy to read.

**Weaknesses:**

1. Although the proposed GazeVQA contains 410K QA pairs, the images are from the GazeFollow dataset, and no additional images are included; therefore, the contribution is somewhat limited.

2. The proposed GazeVQA is evaluated on a limited number of VLMs, making the evaluation insufficient. The authors should conduct evaluations on a wider range of VLMs and provide a comprehensive analysis.

3. The scope of the VQA tasks is somewhat limited; only four types of questions are proposed.

**Questions:**

1. What is the rationale for using BLEU and ROUGE-L to evaluate gaze-point predictions? Why not use an LLM as a judge?

2. How can hallucinations in VLLMs be avoided in Step I—generating the referring expression?

---

### Official Review · Reviewer_8dbc · 2025-10-27

**Soundness:** 3
**Presentation:** 3
**Contribution:** 3
**Rating:** 4
**Confidence:** 4

**Summary:**

This work introduces GazeVQA, the first large-scale benchmark designed to explore the potential of Visual Language Models (VLM) in gaze tracking tasks. This dataset reframes the gaze-following task as a Visual Question Answering (VQA) problem.

**Strengths:**

This paper makes a valuable contribution to bridging gaze following and VLM research, with notable strengths across originality, quality, clarity, and significance. Below is a detailed assessment of each dimension.

1.The innovation of this paper lies in establishing a novel gaze tracking paradigm. Traditional gaze tracking methods rely on coordinate regression, which lacks intuitiveness for users. This paper redefines the task as a Visual Question Answering (VQA) problem, an innovative perspective aligned with human gaze expression habits.

2.This paper introduces GazeVQA, the first large-scale gaze tracking benchmark based on Visual Question Answering (VQA). Unlike existing datasets that provide only bounding boxes or coordinate annotations, GazeVQA offers textual annotations for both observers and gaze targets, covering four practical query types. This fills a gap in the gaze tracking field by enabling fine-tuning of visual-language models and introducing language-aware supervision.

**Weaknesses:**

While the paper makes valuable contributions to gaze following and VLM research, it exhibits several notable weaknesses that limit its depth, generalizability, and explanatory power. Below is a detailed analysis of these flaws.

1.Over-reliance on a single foundational dataset limits universality. The GazeVQA benchmark is constructed solely based on the GazeFollow dataset, and this narrow foundation results in insufficiently diverse scenarios within the dataset.

2.The paper notes that existing VLMs (e.g., GPT-4o, Qwen2.5-VL) have “limited zero-shot performance” but provides only cursory explanations of why these failures occur. Omits qualitative case studies of representative failures. For example, the paper references Figure 2 for zero-shot performance but does not analyze specific examples to identify root causes .

**Questions:**

1. Please clarify the explicit rationale for selecting only GazeFollow as the foundational dataset.

2. Expand the sources and application scenarios of the dataset.

---

### Official Review · Reviewer_ZTCZ · 2025-10-28

**Soundness:** 2
**Presentation:** 4
**Contribution:** 3
**Rating:** 6
**Confidence:** 3

**Summary:**

This paper reframes gaze following from coordinate regression to a VQA paradigm and introduces GazeVQA, a large-scale benchmark comprising 410K QA pairs over 102K images.  The dataset is built via a VLM-powered pipeline that auto-generates observer referring expressions and gaze-point descriptions using structured prompts.

**Strengths:**

(1) Casting gaze following as natural-language VQA is well-motivated, more user-friendly, and aligns with downstream reasoning and interaction.
(2) GazeVQA provides richly structured, diverse supervision (four task types) and explicitly models refusal cases for robustness.
(3) Substantial improvements after supervised fine-tuning, including cross-dataset generalization.

**Weaknesses:**

(1) Although the paper introduces a novel perspective by transforming the gaze-following task from traditional coordinate regression to a VQA paradigm, this innovation may not significantly distinguish itself from existing VLM research. While converting coordinate prediction into a language-based output is interesting, the practical benefits of this approach are not convincingly demonstrated. For instance, is the language-based gaze target description truly more advantageous than direct spatial coordinates in real-world applications?
(2) Among the four task types, the "Refuse to Answer" task is designed to improve model robustness, but its practical significance and real-world necessity are not well-discussed. For example, does this module negatively impact the accuracy of other tasks? And is refusal to answer a common or essential requirement in real-world applications?
(3) Limited comparison to coordinate SOTA. Although Type 3 reports coordinate regression, there is no head-to-head comparison with strong coordinate-based gaze-following methods on identical splits/metrics. This makes it hard to position the approach against established baselines.
(4) BLEU/ROUGE may not reflect whether the described target matches the ground-truth point, semantic or grounding-aware metrics (e.g., ACC) are absent.

**Questions:**

(1) Since the dataset is entirely constructed from existing gaze-following datasets and automatically generated textual descriptions, does it lack sufficient diversity to represent real-world user scenarios? Specifically, in complex social interactions (e.g., multi-person or occlusion scenarios), are the automatically generated QA pairs reliable and accurate enough?
(2) In real-world applications, how frequently does the need for the model to refuse to answer arise? Is this task truly necessary? If tasks like "Describing the Gaze Point" already handle ambiguous cases through "no answer," does the "Refuse to Answer" task become redundant?
(3) The paper includes coordinate regression (Type 3) as a task to complement traditional methods. However, if the ultimate goal is to provide language-based descriptions, does this task lose its practical significance? Especially when Type 1 and Type 2 already describe gaze information, is it necessary to retain coordinate regression?

---

### Official Review · Reviewer_auz4 · 2025-11-02

**Soundness:** 2
**Presentation:** 3
**Contribution:** 2
**Rating:** 4
**Confidence:** 3

**Summary:**

This paper introduces GazeVQA, a large-scale benchmark that reformulates gaze following as a vision–language question answering task rather than the traditional approach of regressing to image coordinates. The main argument is that gaze understanding should be expressed in natural language (e.g., “What is the boy on the skateboard looking at?” → “the skateboard deck beneath him”) instead of just outputting a pixel location, which can be unintuitive, fragile, and often ambiguous.

To build the benchmark, the authors take existing gaze-following datasets and augment them with question–answer pairs. They use available metadata from those datasets — such as head location and gaze point — to construct prompts, and then ask a large language model to generate descriptive questions and answers. They then show that training a vision-language model on this data improves its ability to perform gaze estimation.

**Strengths:**

the authors introduce a new large-scale benchmark, GazeVQA, built in a scalable way by leveraging existing gaze datasets and automatically generating question–answer pairs using metadata like head position and gaze point.  Importantly, the benchmark is designed to cover multiple aspects of gaze understanding — describing the target, giving direction, predicting coordinates, and deciding when not to answer — which makes it relevant for real-world human–AI interaction.
the authors show that fine-tuning medium-sized vision–language models on GazeVQA leads to clear improvements in gaze estimation, even surpassing much larger general-purpose models that have not been trained on this data.,The paper is also clearly written, includes thorough evaluations and ablations, and is easy to follow.

**Weaknesses:**

while the benchmark itself is framed as new, the way it is built — using a vision-language model to automatically generate descriptions and QA pairs from existing gaze datasets — is not novel, and the paper does not clearly show what is fundamentally new about that pipeline beyond applying it to gaze.

The authors do not provide convincing evidence that the generated annotations are reliable: there is no real audit of accuracy, bias, or hallucination in the referring expressions or gaze descriptions, so it’s unclear how noisy or sensitive the data may be. hence it is not clear that low performance of other models is due to actual low performance or, in accuracies in datasset.

Although the paper argues that direct coordinate regression is an unnatural way to evaluate gaze, it still includes a coordinate prediction task in the benchmark through the VLM, rather than committing fully to more semantically meaningful outputs like robust gaze direction or relational grounding; that weakens the push that “language answers are better than pixels.”

**Questions:**

1.	From my understanding of the paper, in Type 3 questions the model is asked to output the gaze point as coordinates (x, y). Was there any evaluation of how accurate these coordinate answers are? Are the VLM-generated coordinates used as ground truth in the benchmark, or do you always use the original human gaze points from the source datasets as ground truth during evaluation?
Related: did you run any human evaluation to measure how often the generated QA pairs are actually correct? For example, is the described “object being looked at” really what the person is looking at, and is the referring expression for the person actually unique and accurate?
2.	How do you ensure that your prompts do not introduce bias (for example, with respect to gender, race, age, or other sensitive attributes)? Did you filter or audit for problematic or sensitive descriptions, such as demographic labeling of minors, stereotypes, or medical/ability-related language?
3.	How do you handle hallucination in the dataset? I understand that including “refuse to answer”–type questions may help discourage hallucinated answers, but did you quantitatively evaluate the hallucination rate in the generated QA pairs?

4.	Could you clarify the statement: “Training only on description-type queries (Type 1) leads to weaker results even on description metrics, showing that diverse question types mutually enhance learning”? Specifically, how was this measured, and why does adding the other question types improve performance on Type 1 itself?

---

### Note · Authors · 2025-11-12

I have read and agree with the venue's withdrawal policy on behalf of myself and my co-authors.